

# Phase state of ambient aerosol linked with water uptake and chemical aging in the Southeastern US

Aki Pajunoja[1], Weiwei Hu[2,3], Yu J. Leong[1,4], Nathan F. Taylor[5], Pasi Miettinen[1], Brett B. Palm[2,3], Santtu
Mikkonen[1], Don R. Collins[5], Jose L. Jimenez[2,3], and Annele Virtanen[1]

[1]Department of Applied Physics, University of Eastern Finland, Kuopio Campus, P.O. Box 1627, 70211 Kuopio, Finland

[2]Cooperative Institute for Research in Environmental Sciences, University of Colorado, Boulder, CO, USA
[3]Department of Chemistry and Biochemistry, University of Colorado, Boulder, CO, USA

[4]Department of Civil and Environmental Engineering, Rice University, 6100 Main St MS-519, Houston, TX, United States

[5]Department of Atmospheric Sciences, Texas A&M University, College Station, TX, USA

*Correspondence to:* A. Virtanen (annele.virtanen@uef.fi)

**Abstract**. During the summer 2013 SOAS field campaign in a rural site in the Southeastern United States, the effect of
hygroscopicity and composition on the phase state of atmospheric aerosol particles dominated by the organic fraction was
studied. The analysis is based on hygroscopicity measurements by a Hygroscopic Tandem Differential Mobility Analyzer
(HTDMA), physical phase state investigations by an Aerosol Bounce Instrument (ABI), and composition measurements using
a high resolution time-of-flight aerosol mass spectrometer (HR-ToF-AMS). To study the effect of atmospheric aging on these
properties an OH-radical oxidation flow reactor (OFR) was used to simulate longer atmospheric aging times of up to three
weeks. Hygroscopicity and bounce behavior of the particles had a clear relationship showing higher bounce at elevated RH
values for less hygroscopic particles which agrees well with earlier laboratory studies. Additional OH oxidation of the aerosol
particles in the OFR increased the O:C and the hygroscopicity resulting in liquefying of the particles at lower RH values. At
the highest OH exposures, the inorganic fraction starts to dominate the bounce process due to production of inorganics and
concurrent loss of organics in the OFR. At typical ambient RH and temperature, organic dominated particles stay mostly liquid
in the atmospheric conditions in the Southeastern US, but they often turn semisolid when dried below ~50% RH in the sampling
inlets. While the liquid phase state suggests solution behavior and instantaneous equilibrium partitioning for the SOA particles
in ambient air, the possible phase change in the drying process highlights the importance of thoroughly considered sampling
techniques of SOA particles.





# 1    Introduction

Atmospheric secondary organic aerosols (SOA) result from gas-phase oxidation of volatile organic compounds (VOC) (Hallquist et al. 2009), which are emitted from anthropogenic and biogenic sources. Chemical aging of the SOA particles in the atmosphere controls their physical and chemical properties such as phase state (Pajunoja et al. 2015), volatility (Kroll et al. 2011) and hygroscopicity (Jimenez et al. 2009, Massoli et al. 2010). Phase state and water uptake properties can also affect the growth of the SOA particles (Riipinen et al. 2012, Shiraiwa and Seinfeld 2012). All these factors can affect the particles' ability to act as cloud condensation nuclei (CCN) and ice nuclei (aerosol-cloud interactions, ACI) or to scatter and absorb solar radiation (aerosol-radiation interaction, ARI) (Stocker et al. 2014).

Several recent studies have shown that SOA particles can be in a semisolid physical phase depending on the particle composition and surrounding humidity conditions (Virtanen et al. 2011, Renbaum-Wolff et al. 2013, Pajunoja et al. 2014, Bateman et al. 2014, Saukko et al. 2015, Pajunoja et al. 2015, Song et al. 2015, Zhang et al. 2015). The phase state is typically represented by the viscosity of the material: materials with viscosities less than $10^2$ Pa s are considered liquids, with $10^2 - 10^{12}$ Pa s as semisolids, and viscosities greater than $10^{12}$ Pa s represent amorphous solid, glassy, material (Koop et al. 2011, Shiraiwa et al. 2011). Material viscosity depends on temperature and relative humidity as particle-phase water can act as a plasticizer (Zobrist et al. 2011) and can decrease the glass transition temperature of the material (Wang et al. 2014). The semisolid or solid phase state of the SOA particles can limit the diffusion of condensable gas-phase molecules from the surface into the particle bulk (Koop et al. 2011, Shiraiwa et al. 2011, Riipinen et al. 2012, Kuwata and Martin 2012, Lienhard et al. 2014). This may affect inner mixing and disturb the equilibrium in gas/particle partitioning and result in slower evaporation of the particles than expected (Vaden et al. 2011, Saleh et al. 2011, Perraud et al. 2012, Abramson et al. 2013). The existence of very low volatility compounds in the particles may also considerably decrease the particle evaporation rates (Cappa and Wilson 2011, Ehn et al. 2014). Transport of small molecules into the particle bulk (e.g. $H_2O$, $O_3$) may not be limited by diffusion in the organic matrix within atmospherically relevant timescales and at temperatures close to room temperature (Price et al. 2014, Pajunoja et al. 2015). However, it has been recently shown that at low temperatures kinetic limitations of water diffusion in organic aerosol particle can affect the ice nucleation processes (Wang et al. 2012, Berkemeier et al. 2014). It should be noted that despite the recent SOA viscosity studies, all the current regional and global aerosol models treat particles as liquid droplets considering no particle phase diffusion limitations.

Several studies have focused on the relationship between the aerosol hygroscopicity (quantified with the hygroscopicity parameter $\kappa$) and degree of oxidation (measured as O:C) showing the correlation between these two factors (e.g. Jimenez et al. 2009, Massoli et al. 2010, Duplissy et al. 2011) while some studies did not show a clear correlation between these parameters (Meyer et al. 2009, Tritscher et al. 2011, Alfarra et al. 2013). There is a lack of studies relating these two factors to physical phase state of the particles. Saukko et al. (2012) showed that the increasing O:C of SOA particles decreases the particle liquefying RH. Recent laboratory studies (Pajunoja et al. 2015) showed that the decrease in the liquefying RH with O:C can be related to the increasing hygroscopicity $\kappa$ measured under subsaturated conditions. They also showed that for laboratory



generated semisolid SOA particles the low water solubility, rather than particle-phase diffusion, restricts the water uptake of SOA. Similar conclusions were drawn by Li et al. (2015). In addition, there are only very few studies on physical phase state of atmospheric aerosols. Virtanen et al. (2010) reported that aerosol particles in boreal forest dominated by SOA can be amorphous solids under dry conditions. More recently, Bateman et al., (2016) showed that in the Amazonian rain forest the

particles are always liquid at relative humidity conditions relevant for that area. Apart from these studies, the information on the physical phase of the atmospheric organic aerosols is scarce and the data showing the relationship of particle physical phase to the O:C and $\kappa$ for ambient aerosol dominated by the organic fraction are lacking.

This study focuses on characterizing the phase state of ambient aerosol particles and on how hygroscopicity and composition of atmospheric particles dominated by organic compounds affects the phase state of the particles. Direct viscosity or diffusion

coefficient measurements are very challenging for ambient conditions where concentrations are low and conditions can be changing relatively rapidly. Instead, we use a robust method based on detection of particle bounce during impaction to infer information about the physical phase of the ambient aerosol particles (Virtanen et al. 2010, Saukko et al. 2012, Bateman et al. 2014, Pajunoja et al. 2015, Bateman et al. 2016). The measurements were performed in a rural environment in the Southern US during the SOAS field campaign in the summer of 2013. In this study we utilize the data from the aerosol bounce instrument

(ABI), a hygroscopic tandem differential mobility analyzer (HTDMA) and a high resolution time-of-flight aerosol mass spectrometer (HR-ToF-AMS) to study the relation of aerosol physical phase state to aerosol composition, O:C of the organic fraction, and particle hygroscopicity.

## 2    Experimental methods

The measurements were conducted in a rural site in Centreville, Alabama, between June 1$^{st}$ and July 15$^{th}$, 2013 during the Southern Aerosol and Oxidant Study (SOAS) (http://soas2013.rutgers.edu) campaign. The southeast US represents an unusual region with a cooling trend compared to the long-term warming trend observed in other parts of the US (Goldstein et al. 2009). The measurement site represents an isoprene- and terpene-rich environment (Guenther et al. 2012) with anthropogenic influence, where the isoprene and terpene SOA and more oxidized organic aerosols are the main organic aerosol (OA)

constituents  (Budisulistiorini et al. 2015, Cerully et al. 2015, Hu et al. 2015, Lee et al. 2016) and the main contributors for the hygroscopicity of the organic fraction (Cerully et al. 2015). The data from ABI, HTDMA and AMS were utilized in the study and the sampling and measurement conditions for each system are described in detail below.

### 2.1    ABI measurements and inferred phase state

ABI was developed to indicate the phase state of SOA particles by measuring the bounced fraction (BF) of the particles

(Pajunoja et al. 2015, Saukko et al. 2012, Virtanen et al. 2011). In this study, the same setup of ABI was employed as in Pajunoja et al. (2015). The ABI was housed in an air conditioned container sampling through a 3m long (above the container)





stainless steel tube with residence time of the sample in the inlet being approx. 5 seconds. In the ABI the particles are dried in a silica gel diffusion dryer to $25 - 30\%$ RH, neutralized with a radioactive $^{210}$Po strip and then size-selected with a differential mobility analyzer (DMA, custom made, open loop sheath flow system). Dried and filtered compressed air was used as sheath air in the DMA with sample-to-sheath flow ratio of 1:5 (i.e. 2 lpm and 10 lpm). The monodisperse aerosol flow passes through

a needle valve for the humidification control unit. The control unit consists of two Nafion humidifiers. The first one (Permapure inc, model: PD-24-SS) is used for humidifying the sheath air of the second Nafion humidifier (Permapure inc, model: PD-240-12SS) which is used for humidifying the sample air. The humidity was varied between ~30-70% in the humidity control unit but the scanning was performed slowly with always less than 10% RH per hour change. To optimize the time resolution of the measurement the RH range of ~30–70% was chosen based on the previous laboratory measurements that showed the biggest

changes in bounce for isoprene and highly oxidized monoterpene SOA occurred in that RH range (Pajunoja et al. 2015). Residence time of the aerosol in the humidified region is approximately eight seconds. The initial humidified monodisperse particle number concentration is measured with a CPC (TSI, 3010) prior to the MOUDI single stage impactor. The impactor plate is covered with polished and cleaned aluminum foil. The number concentration in the output flow of the impactor is measured by another CPC (TSI, 3010) and the bounced fraction (BF) is calculated based on the ratio of these two CPC readings.

The calculations are described elsewhere in more detail (Pajunoja et al. 2015). The upstream pressure was adjusted to $850\pm10$ mbar while the downstream pressure was kept at $700\pm10$ mbar. This pressure difference leads to a cut-off size of the impactor stage of 67 nm in aerodynamic size. The impactor stage was calibrated with dioctyl sebacate (DOS, Sigma-Aldrich) oil particles prior to and after the campaign. In this study, 100 nm particles (electrical mobility diameter) were selected to make sure that the aerodynamic sizes are clearly higher than the cut-off of the impactor stage. RH sensors (Vaisala, HMP-110) in

the ABI were tested with bounce behavior of pure ammonium sulfate particles before and after the campaign and the offset between the two CPCs (i.e. $BF_{reference}$, $\Delta BF_{reference} = \pm0.02$) was measured every second day.

The phase state of the submicron OA particles is inferred based on the ABI measurements. Based on laboratory measurements the particles can be classified as semisolid and liquid particles based on their bounce behavior; the particles having $BF > 0.8$ are semisolid or solid (nonliquid) and for the particles having $BF < 0.1$ the phase transition to liquid has been complete. When

BF is between 0.1–0.8 the particles are in the range of phase transition. A similar classification was introduced and used previously (Saukko et al. 2012, Bateman et al. 2014, Pajunoja et al. 2015) but a simple relationship between BF and viscosity has not been published. Pajunoja et al. (2015) and Li et al., (2015) also used the transition RH from nonliquid to liquid (i.e. RH where BF reaches ~0, liquefying RH) as a parameter representing phase state of the SOA particles. The classification requires particles to be composed of an amorphous phase material. To focus especially on the properties of organic fraction of

the particles, only the data points where the mass fraction of OA is $f_{OA} \geq 0.6$ have been taken into account in the analyses below. However, as aerosol particles are oxidized further by an OH-OFR, $f_{OA}$ can decrease remarkably and thus inorganics may start to control the bounce behavior (see sect. 3.2.).





## 2.2 HR-ToF-AMS measurements and chemical composition

Non-refractory submicron particle-phase masses were measured by an Aerodyne High Resolution Time-of-Flight Aerosol Mass Spectrometer (HR-ToF-AMS, hereafter called AMS) to quantify OA, sulfate, nitrate, ammonium, and chloride (DeCarlo et al. 2006). The elemental compositions (oxygen to carbon ratio, O:C and hydrogen to carbon ratio, H:C) of total OA were

quantified from the high-resolution spectra using the updated ambient calibrations of Canagaratna et al., (2015). Detailed information on AMS analysis and measurements can be found in Hu et al. (2015).

Organic and inorganic fractions of particles were calculated based on AMS analysis (Canagaratna et al. 2007). The composition of $PM_1$ was assumed to represent the average composition of the particle sizes selected in ABI and HTDMA for the analysis. The measured aerosol material can be divided into inorganic ions ($[NH_4^+]$, $[SO_4^{2-}]$, $[Cl^-]$, $[NO_3^-]$) and organic ($[OA]$) mass

concentrations. Actual inorganic salts present in the particle phase are then estimated based on the method introduced by Nenes et al., (1998). In the analysis the inorganic aerosol is divided into 3 categories based on a molar ratio of ammonium ions to sulfate ions, $R_{SO_4} = [NH_4^+] / [SO_4^{2-}]$ according to the following classifications: 1) if $R_{SO_4} < 1$, sulfate exists as a mixture of sulfuric acid (SA) and ammonium bisulfate (ABS), 2) if $1 < R_{SO_4} < 2$, sulfate is in form of ammonium sulfate (AS) and ABS, and 3) if $R_{SO_4} > 2$, sulfate is existing as AS (Nenes et al. 1998, Cerully et al. 2015). The classification assumes the contribution

of nitrates (<1 % of total $PM_1$ in SOAS) and other inorganic species (e.g. sodium and chloride) to be negligible in the particle phase. Most aerosol nitrates were organic during SOAS (Lee et al. 2016). After the classification, species volume fractions are estimated from the mass concentrations assuming densities of the species listed in Table 1.

## 2.3 HTDMA measurements and hygroscopicity analysis

An HTDMA was used to measure water uptake of ambient aerosol at subsaturated conditions (Brechtel and Kreidenweis 2000).

Initial sample air was dried with a Nafion drier to RH<30% and then neutralized with a bipolar $Po^{210}$ charger. After that a monodisperse aerosol distribution was selected with the first DMA (DMA1, high flow, custom made, closed loop sheath flow system). Then the selected particle population was exposed to a high, fixed RH=90±3% and entered the 2nd DMA (DMA2) that was operated as a scanning mobility particle sizer (SMPS). The residence time of the particles in the humidified conditions was about eight seconds. The DMA2 was controlled with an adjustable voltage control (0–10kV) and particle number

concentration was measured with a CPC as a function of DMA2 voltage (scanning time 3 min).

The hygroscopic growth factors (GF, the ratio between humidified and dry particle diameter) were calculated at actual RH and only the data when RH=90±3% were analyzed. The GF data were corrected to fixed RH=90%. Probability density functions of the HTDMA measurements (GF-PDF) were evaluated by using the TDMAinv inversion toolkit (Gysel et al. 2009). In the analysis of ABI, HTDMA and AMS data only the time periods corresponding to unimodal GF-PDF curves were taken into

account to prevent including data from externally mixed aerosols in the analysis.

Hygroscopicity-$\kappa$ was calculated by using (RH, GF)-data according to equation (11) in Petters and Kreidenweis (2007):




$$\kappa_{tot} = 1 - GF^3 + \frac{GF^3-1}{\frac{RH}{100\%}} e^{\left(\frac{4\sigma_W M_W}{RT\rho_W d_p GF}\right)} \tag{1}$$

with size-selected dry diameter of the particles $d_p$, surface tension of water $\sigma_w$=0.072J/m², temperature $T$=297K, molecular weight of water $M_w$=0.018kg/mol, gas constant R=8.3145Jmol⁻¹K⁻¹, density of water $\rho_w$=1000kg/m².

The hygroscopicity of the organic fraction ($\kappa_{OA}$) was calculated from total hygroscopicity ($\kappa_{tot}$) measured by the HTDMA using a $\kappa$-mixing rule (Petters and Kreidenweis 2007). In general, hygroscopicity of multicomponent aerosol particles can be estimated as a sum of contributions of each component (Petters and Kreidenweis 2007). Thus, the mixing rule (Eq. (2)) can be separated into the organic (OA) fraction and inorganic (inorg) part (Eq. (3)) and the inorganic part can be divided further as (Eq. (4)):

$$\kappa_{tot} = \sum_{i=1}^{n} f_i \kappa_i \tag{2}$$

$$= f_{OA}\kappa_{OA} + f_{inorg}\kappa_{inorg} \tag{3}$$

$$= f_{OA}\kappa_{OA} + f_{AS}\kappa_{AS} + f_{ABS}\kappa_{ABS} + f_{SA}\kappa_{SA} + f_{AN}\kappa_{AN} \tag{4}$$

where $f_i$ and $\kappa_i$ are the volume fraction and hygroscopicity parameter of component $i$, respectively (Petters and Kreidenweis 2007). The volume fractions of organic and inorganic species, i.e. AS, ABS, SA and ammonium nitrate (AN), are derived from AMS mass fractions using densities shown in Table 1. The hygroscopicities of inorganic species were calculated with the E-AIM model (see Table 1) (Clegg et al. 1998). The residual water associated with SA at dry conditions (i.e. conditions after the diffusion dryer) was taken into account in the data analysis and was calculated with the E-AIM as well.

## 2.4 Potential aerosol mass oxidation flow reactor (OFR)

An Oxidation Flow Reactor (OFR) was used during SOAS to investigate OA formation/aging from ambient air on the phase state of the particle over a wide range of OH exposures ($10^{10}$–$10^{13}$ molec. cm⁻³ s). The OFR is a cylindrical vessel (~13 L) with flow of 3.5–4.2 lpm and gas-phase residence time of approximately 200 seconds. RH and T inside the OFR were not actively controlled but rather determined by the ambient air sampled without drying. The OFR was mounted on the roof of a measurement trailer.

OH radicals are generated when the UV lights initiate $O_2$, $H_2O$, and $O_3$ photochemistry. The variation of OH radical concentration was achieved by varying the voltage of two UV lamps, which were mounted in Teflon tubes inside the OFR. For the conditions of the SOAS study, oxidation has been shown to be dominated by OH, while non-tropospheric chemistry




in the reactor is negligible (Peng et al. 2015). The OH exposure ($OH_{exp}$) was calculated by fitting the parameters in the equation (5) reported in Li et al., (2015) to OH exposures calculated by the real-time decay of CO injected into the OFR (1–2 ppm).

$$OH_{exp} = A \cdot O_3{}^P \cdot H_2O \qquad\qquad (5)$$

During the last two weeks of the campaign, the ABI and AMS were sampling through the OFR whereas the HTDMA was measuring ambient air throughout the campaign. Both AMS and ABI had separate valve systems to multiplex sampling between OFR and bypass in 10 minute steps. The sampling line between the OFR and ABI was approx. 12m long copper tubing whereas all the other inlet lines (i.e. for HTDMA, AMS and ABI bypass) were approx. 3m long. However, the tubing diameters were chosen such that the residence time in both tubings were approx. 5 seconds.

### 2.5 Description of Structural Equation Modeling (SEM)

SEM is a series of statistical methods that allow complex relationships between one or more independent variables and one or more dependent variables (See e.g. Kline 2015). SEM is most commonly thought of as a combination of factor analysis, multiple regression analysis and analysis of variance (ANOVA). It is used to analyze the structural relationship between measured variables and latent constructs and it estimates the multiple and interrelated dependence in a single analysis. It can be remarked that the SEM allows one to perform some type of multilevel regression/ANOVA on factors. The SEM model used in this study can be categorized as path analysis or more specifically model of causal interference (Pearl 2009). Causal models are usually presented as a directed acyclic graphs, where the nodes represent the variables and the edges represent the causal relationship so that the arrow shows the direction of the effect (Karvanen 2015). A graphical model visualizes the causal relationships, and is a mathematically well-defined object from where causal conclusions can be drawn in a systematic way. The model in this study is built to describe the factors affecting BF in the measurement data and their internal relationships. The analyses were performed with PROC CALIS in SAS 9.4 software (SAS Institute, Cary NC).

### 3 Results

The time series of particle hygroscopicity and total particulate mass ($PM_1$) and composition measured by HTDMA and AMS respectively as well as weather parameters ($T$, RH, radiation) measured during the SOAS campaign are shown in Fig. 1. All the data is averaged over 10 minute intervals and time periods (shorter than 1 hour) where data is missing are linearly interpolated. The periods where precipitation was more than 3 mm over 3 hours have been removed from the data analysis. $T$, RH and $PM_1$ are shown in Fig. 1a. The background color of the Fig. 1a represents solar radiation measured at ground level at the site. Both $T$ (red line in Fig. 1a) and RH (blue line) were relatively high during the campaign and have strong diurnal variations as expected. Total $PM_1$ (black line) was in the range of 1–10 µg m$^{-3}$ most of the time. The time series of composition of the particles calculated as described in sect. 2.2. is shown in Fig. 1b and the overall mean values are presented in a pie chart





next to the plot. OA (mean mass fraction 67%) dominated the $PM_1$ but especially the contribution of ammonium bisulfate (ABS, mean mass fraction 25%) increased at times. Apart from a few mornings the estimated SA concentration was low and had a campaign mean value of 7%. The contribution of AS was low during the entire campaign (< 1%). According to the studies by Kim et al. (2015), Hu et al., (2015) and Xu et al. (2015) OA is dominated by isoprene and monoterpene SOA, with

smaller contributions from anthropogenic and biomass burning sources.

The measured inorganic and organic fractions and the $\kappa$-mixing rule (see Eq. (4)) are used to estimate hygroscopicity for organic fraction of the particles, $\kappa_{OA}$. Total hygroscopicity, $\kappa_{tot}$, derived from HTDMA measurements at RH = 90±3% is shown in Fig. 1c. $\kappa_{tot}$ is in the range of 0.1–0.4 whereas $\kappa_{OA}$ varies between 0–0.3. As expected, the $\kappa_{tot}$ is highest when the fraction of organics is lowest. Nevertheless, increasing $\kappa_{tot}$ cannot be explained only by the increased fraction of inorganics as

the $\kappa_{OA}$ also peaked at the same time implying a more hygroscopic OA (Jimenez et al. 2009, Massoli et al. 2010). $\kappa_{OA}$ and $\kappa_{tot}$ are consistent with other studies conducted during the SOAS campaign in southeast US (Nguyen et al. 2014, Cerully et al. 2015, Brock et al. 2015).

Diurnal profiles of key parameters are shown in Fig. 2. The data shown in Fig. 2 is averaged over 30 min periods. Quartiles ($q_{25\%}$ and $q_{75\%}$) of the 30 minute averages are added to Fig. 2a with shaded color. Both $T$ and RH have typical diurnal profiles:

RH drops in the early afternoon, when $T$ reaches its daily maximum. At nighttime $T$ stays typically above 20°C while RH is near the saturation point. $PM_1$ (Fig. 2a) is rather constant during the day and night. In Figure 2b and 2c, the median values of $f_{OA}$ and O:C are shown. Hygroscopicity growth factor (GF at 90%RH) data is also averaged over 30 min intervals and is shown in Fig. 2d. The diurnal profile of OA fraction has an opposite trend to that of the GF distributions; the mean GF increases as $f_{OA}$ decreases, which is not surprising as generally the atmospheric inorganic species are more hygroscopic than SOA (Petters

and Kreidenweis 2007). On the other hand, the diurnal profiles of GF and O:C showed similar trends; at nighttime the mean GF was at the lowest range whereas at daytime they increase together. This also suggests that the increase in total hygroscopicity in daytime partly resulted from increased oxidation of the OA, and not only from increased inorganic fraction. However, the O:C of the particles peaked in the afternoon but its diurnal profile does not follow exactly the same profile as the GF distributions which reach a maximum approx. 2 hours earlier.

The measured BF plotted as a function of impactor RH of the ABI is shown in Fig. 3. Generally, the data spreads over a relatively wide range of BF and the variation of BF depends strongly on RH. At RH < 40% all measured values show high bounce indicating the a semisolid or solid phase state of the particles. No evidence of liquid particles at these relatively dry conditions was found. As RH increases, the scatter in BF(RH) increases due to the variations in particle composition. The relationship between the composition and bounce behavior is discussed in more detail in section 3.2. Generally, the bounce

started to decrease at RH values approaching 40–60% at the latest indicating an early stage of the transition from semisolid to liquid phase. Frequency histograms of ambient T and ambient RH are added to the Fig. 3. as a subplot. When considering the ambient humidity conditions at the measurement site (see subplot in Fig. 3; RH varying mostly between 70–100%), it can be concluded that organic dominated aerosol particles are mostly in the liquid phase during the summer season in the southeast





US. However, as ambient RH typically drops during afternoons and if the OA fraction is high enough, the particles can be in a semi-solid phase in ambient.

As discussed above, the OA at the SOAS site is dominated by isoprene- and monoterpene-SOA. Results from Pajunoja et al. (2015) for SOA stemming from ozonolysis of α-pinene and isoprene are added to Fig. 3 to illustrate a typical bounce behavior of pure laboratory SOA. As can be seen, the major fraction of the ambient measurements falls between the BF curves measured for isoprene SOA with O:C 1.07 and monoterpene (α-pinene derived) SOA with O:C 0.56 showing that the ambient observations are consistent with the laboratory studies.

### 3.1 Link between hygroscopicity, oxidation level and phase state

To investigate how the particle phase state, hygroscopicity and degree of oxidation are linked, we combined the data measured by ABI, HTDMA and AMS. Bounce behavior is sensitive to RH in the range where the particles undergo humidity induced phase transition as water acts as a plasticizer. For this reason, even small changes in RH cannot be ignored in the analysis. To analyze the phase change and factors affecting it, we plot (RH, BF)-scatterplots where the values of $f_{OA}$, $\kappa_{tot}$, O:C, and $\kappa_{OA}$ are represented by color coding (Fig. 4). The (RH, BF)-area is divided into pixels, one pixel being 0.5%RH wide and 0.02 high. The value of each pixel represents the average of all the data in the pixel. To isolate the effect of OA fraction on the phase state and to exclude the rainy periods we have filtered the data to include only $f_{OA} \geq 0.6$ and precipitation < 3mm 3h$^{-1}$. The same filter is used also in section 3.2. All particles behave like solids/semisolids at RH < 40% but as RH increases, BF drops to an extent that depends on the chemical composition and hygroscopicity of the particles. Typically, the particles with the highest organic fraction (see Fig. 4a, dark red, $f_{OA} > 0.8$) bounce even at high RH (range of RH) indicating semisolid phase of the OA fraction. At the same time $\kappa_{tot}$, O:C and $\kappa_{OA}$ have the lowest values (see Fig. 4b-d). Hence the observation is in agreement with laboratory results showing that increase in O:C of SOA particles decreases the particle liquefying RH (Pajunoja et al. 2015, Saukko et al. 2012). This is due to the low water uptake of the less oxidized organic material in the particles.

Saukko et al. (2015) showed that in the case of mixed ammonium sulfate and α-pinene derived OA with relatively low O:C and thus separated phases of OA and inorganic fractions, the particle bounce was greatly affected by the organic fraction when the organic mass fraction was ≥ 0.7. In their study, it was shown that the particle bounce started to increase with solidification of the organic fraction even if the inorganic core was deliquesced. This leads to the conclusion, that the bounce behavior of mixed particles is dominated by organic material when $f_{OA} \geq 0.7$. Most of the data points existing below the BF curve of isoprene-SOA in Fig. 3. (blue diamonds) were dominated by inorganics and thus are not represented in Fig. 4a due to the filtering of the data with $f_{OA} \geq 0.6$.

When the most hygroscopic particles are investigated (red area in Fig. 4b), the phase transition from semisolids to liquids starts already at RH = 40–50%. From Fig. 4a-c it can be seen that particles having the highest hygroscopicity and also the lowest bounce at elevated humidity have the highest inorganic fraction and also the highest O:C of the organic fraction resulting in the highest $\kappa_{OA}$. Hence, the high values of $\kappa_{tot}$ results from the high fraction of inorganics and also high $\kappa_{OA}$ and the phase





change at relatively low humidities is driven by the water uptake of both organic and inorganic fractions. The inorganic fraction consists mostly of ABS having DRH and ERH values lower than 40%RH (see Table 1). This allows us to assume that the inorganic fraction is always deliquesced at RH used in the ABI.

Based on the data presented in Fig. 4, we can conclude that the particle phase state is affected both by the contribution of inorganic fraction and the organic fraction. The SEM analysis also supports the conclusions made based on Fig. 4. SEM was used to investigate relationships between (RH, BF) –pair and the key variables shown in Fig. 4. In addition to the key variables, ambient $T$ ($T_{ambient}$) and ambient RH ($RH_{ambient}$) were added to the model since they affect the results mainly by affecting the inlet conditions of ABI (and thus affecting the fixed RH inside the ABI). The SEM results of the ambient data shown in Fig. 4 are listed in Tables 2 and S1 (see Supplementary Information, SI). Table 2 shows the total, direct and indirect effects of the predictor variables on BF and Table S1 describes the magnitudes of individual parameter estimates within the pathway model. Indirect effect can be described such that when e.g. $\kappa_{OA}$ has a significant effect on $\kappa_{tot}$ (see Table S1), which in turn affects BF, the $\kappa_{OA}$ has an "effect pathway" to BF via $\kappa_{tot}$, and magnitude of this effect can be quantified into the indirect effect shown in Table 2. Table 2 shows that RH and $\kappa_{tot}$ have direct effect to BF while $\kappa_{OA}$ and $f_{OA}$ have indirect effects via $\kappa_{tot}$. O:C has both, direct and indirect (via $\kappa_{OA}$) effect on BF. $RH_{ambient}$ can be interpreted as an indicator of local meteorology and diurnal variation. $T_{ambient}$ in turn has an effect on RH and thus has an indirect effect on BF. Based on the data listed in Tables 2 and S1 we can conclude that $\kappa_{tot}$ has the biggest influence on BF, and that $f_{OA}$ is affecting BF indirectly via hygroscopicities.

## 3.2    Effect of particle oxidation in OH-OFR on bounce behavior

The degree of oxidation of ambient OA particles can be enhanced with an OH-OFR flow reactor. The reactor was used for two weeks (from July 1[st] to July 15[th]) to study the effect of the increased oxidation of ambient particles on their bounce behavior. Again, only the cases where the organic fraction of initial ambient aerosol (measured via the bypass) was greater than 0.6, were considered in the analysis. In Figure 5, the measured bounced fraction as a function of relative humidity of the humidification system is depicted. The data is colored by OH exposure (Fig. 5a), OA fraction (Fig. 5b), O:C of the organics (Fig. 5c) and fraction of ammonium bisulfate (Fig. 5d). In general, ABS formed the major fraction of the inorganics.

During the measurement period the O:C of ambient aerosol was 0.5–0.9 for the analyzed cases. When the aerosol was exposed to increased OH concentration in the OFR the $f_{OA}$ as well as the O:C increased when the OH exposure was small. With the very high OH exposure values (OH exposure $> 3–10 \cdot 10^{12}$ molecules cm$^{-3}$ s$^{-1}$) O:C reached very high values (maximum approx. 1.8) but at the same time $f_{OA}$ decreased. These data points are marked in red color in Figs. 5a, c and d and in blue color in Fig. 5b. The increase of the O:C up to very high values and simultaneous decrease of OA fraction is probably due to the strong heterogeneous fragmentation/volatilization of the particulate organic molecules (Ortega et al. 2015, Palm et al. 2016).

When the O:C of the particles exposed to elevated OH concentration was at the same level as the O:C of ambient particles (see Fig. 4a-d and Fig. 6) the bounce behavior was very similar for both ambient and OH aged aerosol. As can be expected when the O:C further increased due to the elevated OH exposure in OFR, the BF measured at certain RH conditions clearly decreased

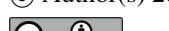



with increasing O:C. The trend is similar to ambient data as can be seen in Fig. 6 where the BF for ambient after OFR treatment and corresponding ambient-only data is presented for selected narrow RH range (48±4%RH) and for particles having $f_{OA}$ greater than 0.6 after the treatment. The data indicates that the highly oxidized particles may undergo an early stage of the phase transition already at RH < 40%.

These conclusions are supported also by the SEM analysis performed for OFR data. Table 2 reports the effects in SEM model built on OFR data and Table S2 (see SI) describes the magnitudes of individual parameter estimates within the pathway model. Table 2 shows that OH concentration has significant decreasing direct and indirect (via O:C and $f_{OA}$) effects on BF. RH and O:C have only direct effects and $f_{ABS}$ has only indirect effect via $f_{OA}$, which has both, direct and indirect (via O:C) effects. $T_{ambient}$ affects again RH but the meteorology-indicating effect of $RH_{ambient}$ on RH is not anymore significant after OFR

treatment. Relationship between BF and O:C is not strong for ambient data but after OFR treatment the relationship between O:C and BF instead is more pronounced, namely that the higher oxidation of OA decreases the bounciness of the particles. For OFR data, $f_{OA}$ has negative and $f_{ABS}$ positive correlation with O:C which is due to loss of organics at high OH-exposures.

## 4    Conclusions

Phase state of ambient particles was inferred from ABI measurements conducted at a rural site in central Alabama between June 1st and July 15th 2013 as a part of SOAS campaign. HTDMA was used to measure water uptake of the dried ambient particles at 90%RH while AMS was employed to resolve dried $PM_1$ composition and O:C of the organic fraction. Organics accounted for 67% of total $PM_1$ during the campaign. Ammonium bisulfate is estimated to be the main contributor of inorganics with mean fraction of 25% while on average only 7% of the particle mass was sulfuric acid.

Based on ABI measurements, we found that the phase transition from semisolid to liquid phase starts in the RH range from 40 to 60% depending on the particle composition. Thus, the organic dominated ambient SOA particles are mostly in the liquid phase in the southeast US at atmospherically relevant summertime RH conditions but they often turn semisolid when dried in the measurement setup. The RH-dependent bounce curves measured for ambient aerosol having an organic mass fraction ≥ 60% falls between the BF curves measured for isoprene and monoterpene SOA generated in the laboratory (Pajunoja et al.,

2015). This is consistent with the findings by Kim et al., (2015), Hu et al. (2015) and Xu et al., (2015) that both types of SOA dominate OA composition at this site. The characteristics of the organic fraction such as O:C and hygroscopicity played a role in the bounce process; particles with the highest organic fraction, lowest hygroscopicity and/or lowest O:C stayed semisolid at higher RH than particles having lower organic fraction and higher hygroscopicity. The statistical SEM analysis revealed that the main factor controlling the liquefying RH of ambient aerosol particles is their hygroscopicity.

Further OH oxidation of SOA by the OFR provides an extended oxidation range for the analysis. The lowest OH exposures increased the organic mass and slightly also the degree of oxidation of OA whereas the higher OH exposures resulted in clearly



higher O:C, a loss of organic mass, and hence an increased mass fraction of ABS. According to the bounce measurements, the increased OH exposure decreased the phase transition RH. This is emphasized at the highest OH exposures.

Based on the measurements we can conclude that in the isoprene and terpene-rich environment influenced also by anthropogenic emissions, the atmospheric aerosols dominated by the organic fraction are mostly in the liquid phase when the temperature and RH conditions are comparable to those in southeast US during the summer season. The increased OH exposure decreased the particle liquefying RH further, implying that any further atmospheric aging of the aerosol does not change the conclusion. The results suggest that in environments similar to the measurement site, it can be assumed that the diffusion limitations in the particle bulk are negligible and the particles can be treated as liquid droplets in regional transport models. The results are in line with the results reported in Amazonia which also represents an isoprene rich environment having elevated RH and temperature conditions (Bateman et al. 2016).

**Acknowledgements**

European Research Council (ERC starting grant 335478), Academy of Finland (259005, 272041), UEF strategic funding. WH, BBP, and JLJ acknowledge funding from NSF AGS-1243354/AGS-1360834, EPRI 10004734, and DOE (BER/ASR) DE-SC0011105. DRC and NFT acknowledge funding from NSF AGS-1242932. BBP acknowledges support from a US EPA STAR Graduate Fellowship (FP-91761701-0). The manuscript has not been reviewed by EPA and thus no endorsement should be inferred.

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





**Table 1.** Literature values of density, molar mass, deliquescence point (DRH), efflorescence point (ERH) and hygroscopicity $\kappa$ for inorganic species. Hygroscopicities of inorganics listed here are based on the E-AIM model (Clegg et al., 1998). Variables written in *italic style* are campaign medians with quartiles in the brackets .

| | Categorized[‡] components of $PM_1$ | | | |
|---|---|---|---|---|
| Abbreviation | AS | ABS | SA | OA |
| Molec. formula | $(NH_4)_2SO_4$ | $NH_4HSO_4$ | $H_2SO_4$ | - |
| Density (kg m$^{-3}$) | 1770 | 1780 | 1840 | 1400 |
| Molar mass (g mol$^{-1}$) | 132.14 | 115.11 | 98.08 | *210 (124/390)* |
| $\kappa_{HGF}$ at 90%RH | 0.45[†] | 0.68[†] | 1.18[††] | *0.10 (0.04/0.16)* |
| DRH (RH%) | 80[*] | 40[**] | - | - |
| ERH (RH%) | 28[*]-36[**] | <22[**] | - | - |
| Volume fraction | *0 (0/1)* | *21 (15/27)* | *3 (1/9)* | *73 (66/79)* |

[‡]Nenes et al. (1998), [*]Smith et al. (2012), [**]Tang et al. (1994), [†,††]E-AIM, [††]residual water ignored





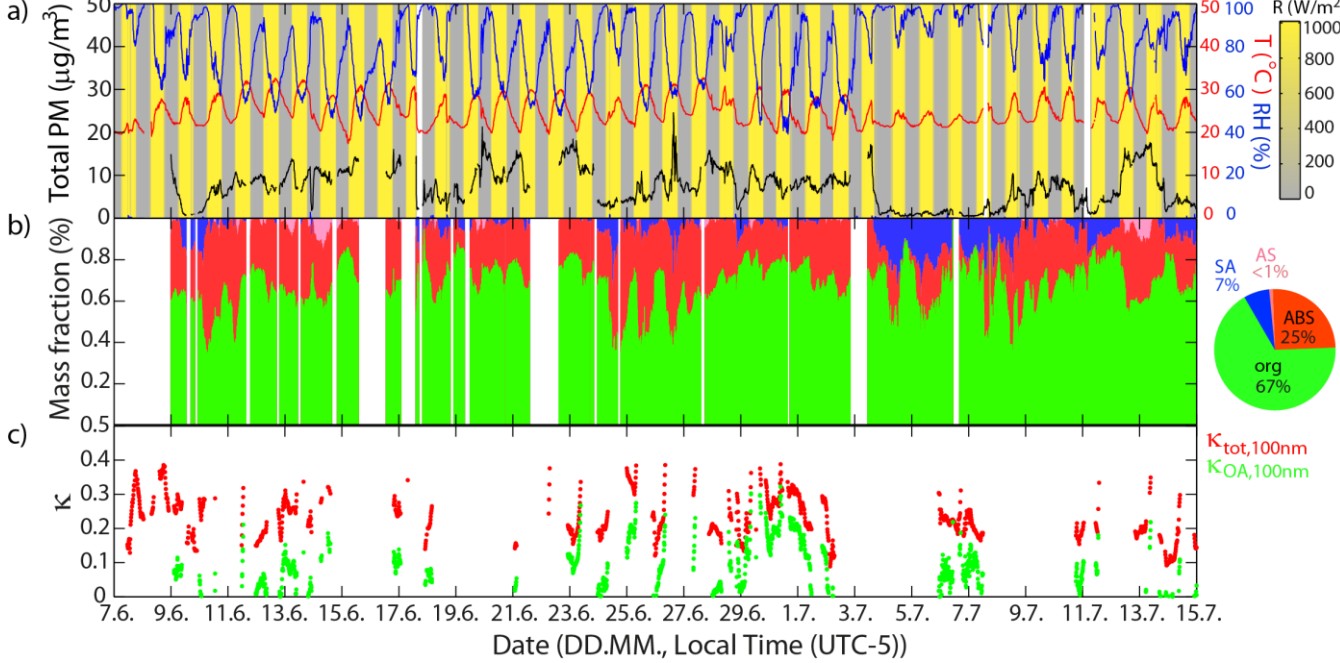

**Figure 1.** Time series of a) ambient temperature, RH, and PM₁ for the entire campaign, b) mass fraction of OA (green, average 67%), and stacked mass fractions of inorganics (blue, pink and red, note that blue represents sulfuric acid instead of the usual nitrate in the AMS color scheme), c) hygroscopicity $\kappa$ of total particles (red dots) and of OA fraction (green dots). Shaded color in a) represents amount of solar radiation (dark bars denote night and yellow bars denote daytime).





**Figure 2.** Diurnal profiles of a) ambient temperature (°C), RH (%), and $PM_1$ (µg/m³), b) mass fraction of organics of the ambient particles ($f_{OA}$), c) degree of oxidation (O:C) and d) median of the hygroscopic growth GF-PDF distributions (GF). Shaded areas in a) and blue boxes in panels b)-c) represent quartiles ($q_{25\%}$ and $q_{75\%}$) of the data which is averaged over 30 min intervals. Black errorbars represent data range calculated based on the distance to the quartiles ($q_{75\%}-1.5(q_{75\%}-q_{25\%}) \leq x \leq q_{25\%}-1.5(q_{75\%}-q_{25\%})$). Data extremes (outside the errorbars) are separated from the data and shown as red crosses.





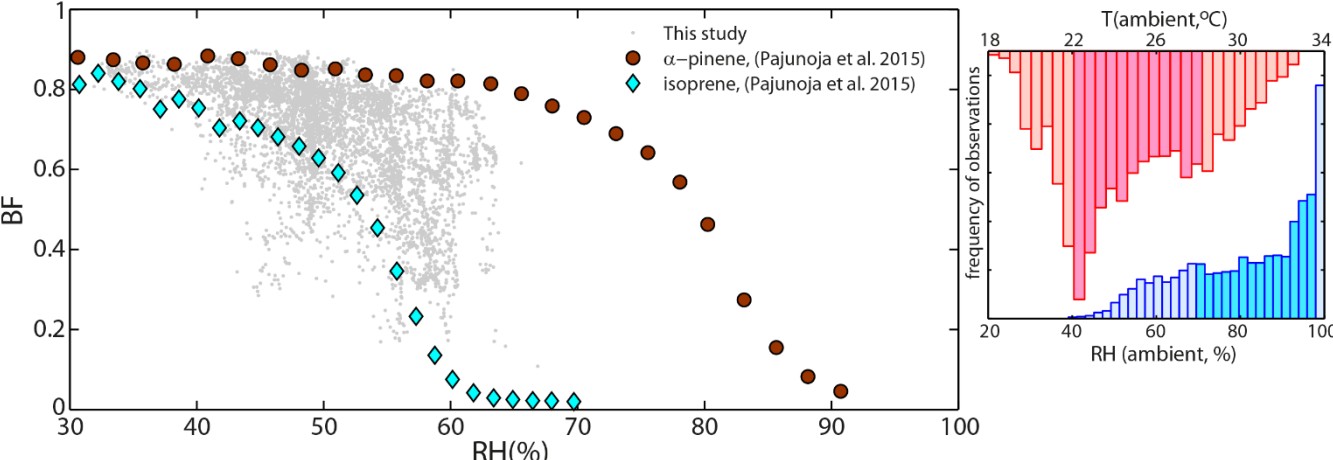

**Figure 3.** Bounced fraction of the ambient SOA particles versus fixed RH in the ABI impactor (left). The bounce data (10min averages) is shown (grey dots) with laboratory results recently reported by Pajunoja at al. (2015) for α-pinene ozonolysis SOA (dark brown circles, O:C 0.56) and isoprene-derived SOA (light blue diamonds, O:C 1.07). Ambient conditions during the campaign ($T$; red bars and RH; blue bars) are shown in the subplot on the right hand side where the darker bars represent data within quartiles ($q_{25\%}$ and $q_{75\%}$). As can be seen, 75% of time of the campaign $RH_{ambient}$ was greater than 70%RH and $T_{ambient} > 22°C$.



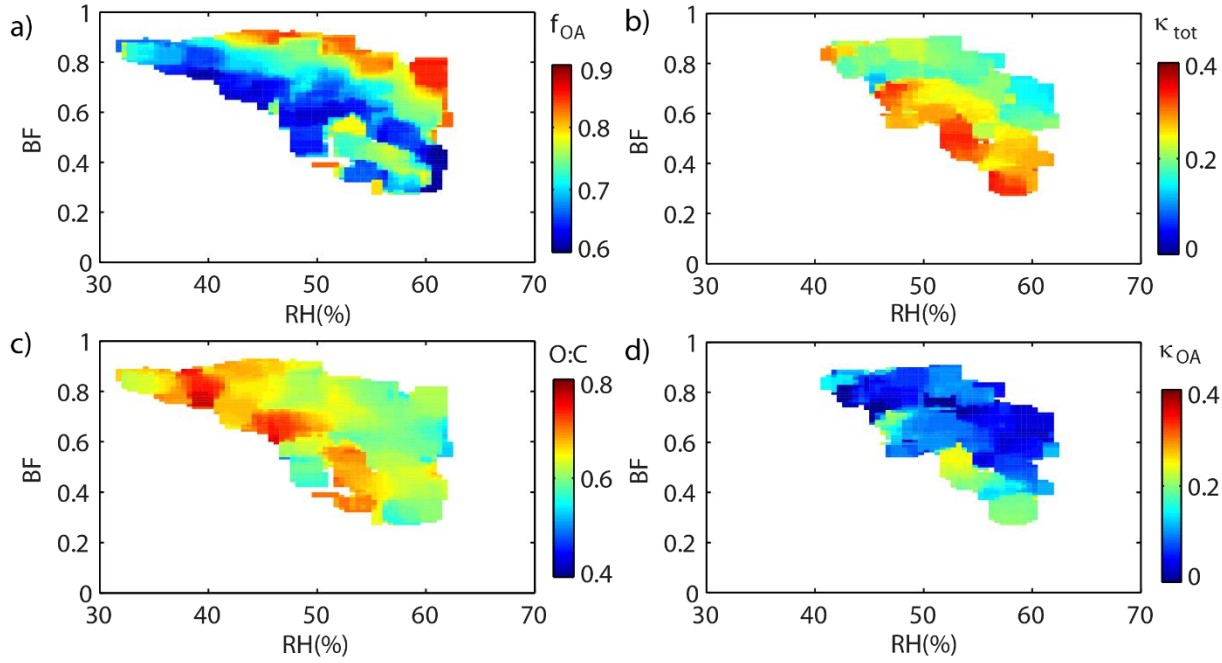

**Figure 4.** Comparison between BF, impactor RH and particle properties for ambient-only data; a) (RH,BF) –data colored with fraction of organics derived from AMS analysis, b) (RH,BF) –data colored with hygroscopicity-$\kappa$ derived from HTDMA results, c) (RH,BF) – data colored with O:C (Canagaratna et al., 2015) derived from AMS analysis, and d) (RH,BF) –data colored with hygroscopicity $\kappa$ of the OA fraction of the particles derived from HTDMA and AMS results. Data is filtered by $f_{OA} \geq 0.6$ and precipitation $< 1$mm h$^{-1}$.





**Table 2.** Standardized effects (direct, indirect and total) on BF based on the SEM model performed for ambient data (Fig. 4.) and OFR data (Fig. 5.). Effects are standardized (i.e. scaled to follow normal distribution with zero mean and variance of unity) in order to compare parameter values that are measured on quite different scales.

| Parameter | | AMBIENT | | | OFR | | |
|---|---|---|---|---|---|---|---|
| | | Total | Direct | Indirect | Total | Direct | Indirect |
| RH | Effect | -0.66 | -0.66 | 0 | -0.5 | -0.5 | 0 |
| | Std Error | 0.02 | 0.02 | | 0.02 | 0.02 | |
| $f_{OA}$ | Effect | 0.14 | 0 | 0.14 | 0.61 | 0.52 | 0.09 |
| | Std Error | 0.02 | | 0.02 | 0.02 | 0.03 | 0.01 |
| O:C | Effect | -0.22 | -0.16 | -0.06 | -0.3 | -0.25 | 0 |
| | Std Error | 0.03 | 0.02 | 0.02 | 0.03 | 0.03 | |
| $T_{ambient}$ | Effect | 0.19 | 0 | 0.19 | 0.24 | 0 | 0.24 |
| | Std Error | 0.03 | | 0.03 | 0.02 | | 0.02 |
| $RH_{ambient}$ | Effect | -0.35 | -0.17 | -0.17 | | n/a | |
| | Std Error | 0.04 | 0.03 | 0.04 | | | |
| $\kappa_{tot}$ | Effect | -0.5 | -0.5 | 0 | | n/a | |
| | Std Error | 0.02 | 0.02 | | | | |
| $\kappa_{OA}$ | Effect | -0.49 | 0 | -0.49 | | n/a | |
| | Std Error | 0.02 | | 0.02 | | | |
| $\log_{10}(OH)$ | Effect | | n/a | | -0.5 | -0.11 | -0.4 |
| | Std Error | | | | 0.03 | 0.02 | 0.02 |
| $f_{ABS}$ | Effect | | n/a | | -0.3 | 0 | -0.25 |
| | Std Error | | | | 0.02 | | 0.02 |



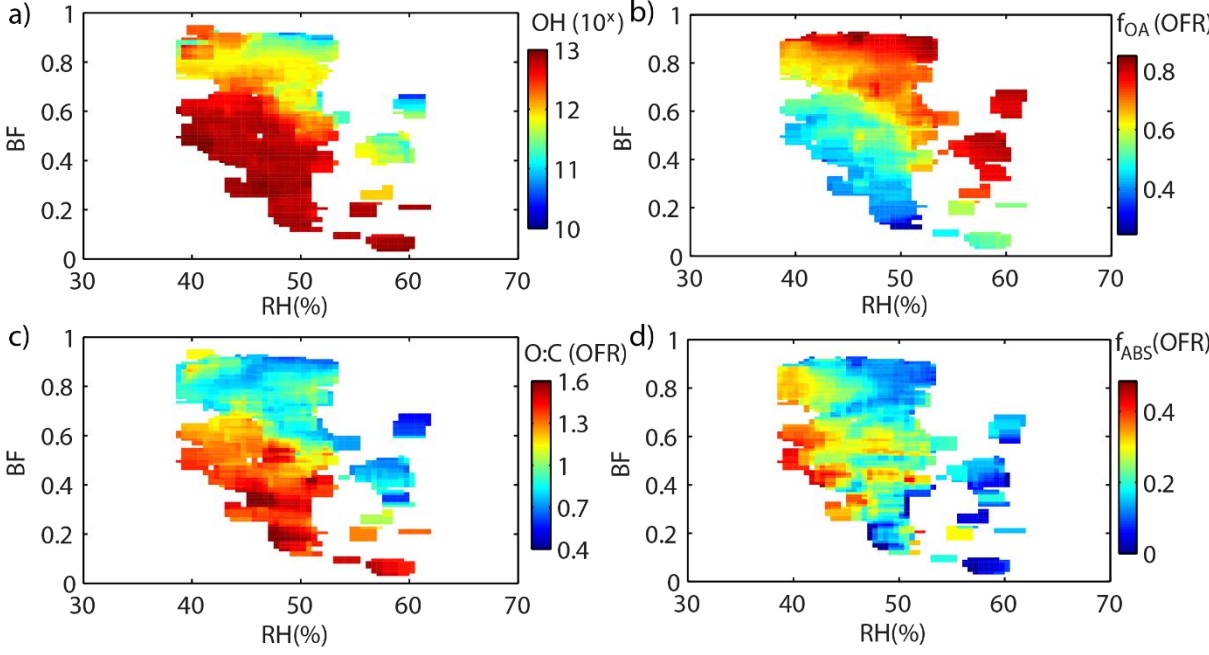

**Figure 5.** Scatterplot between BF, impactor RH and particle properties measured through the OFR; a) (RH,BF) –data colored with exposed OH concentration ($10^x$ molec. $cm^{-3}$ s), b) (RH,BF) –data colored with organic mass fraction of the particles derived from AMS results, c) (RH,BF) –data colored with O:C derived from AMS analysis, d) (RH,BF) –data colored with mass fraction of ammonium-bisulfate derived from AMS analysis and composition categorization as in Cerully et al. (2015). Data is filtered by precipitation < 1mm $h^{-1}$ and ambient aerosols $f_{OA} \geq 0.6$.





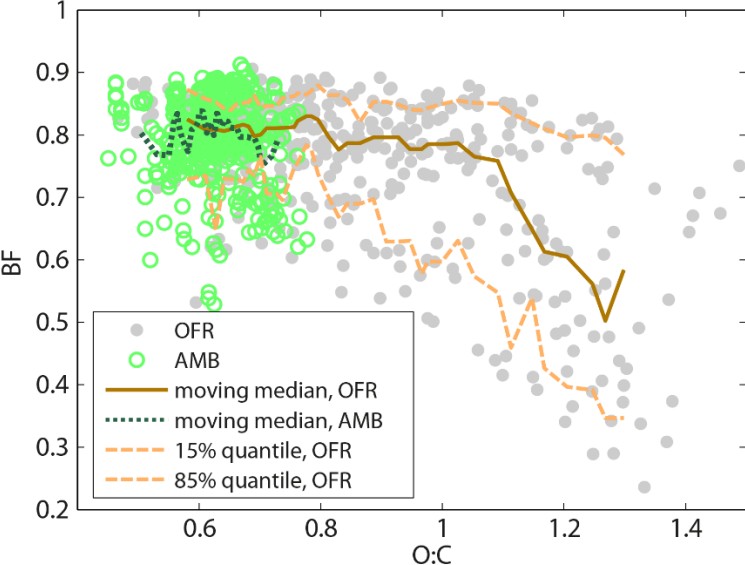

**Figure 6.** Bounced fraction (BF, 10min averages) of the ambient OA (green hollow dots) and corresponding OFR-treated OA (grey dots) versus degree of oxidation (O:C). RH range of the ABI is limited to RH = 48±4% and only data points where $f_{OA} \geq 0.6$ after the OFR treatment has used in the analysis. This has done to minimize the effect of RH and inorganics on BF in the figure. Moving median for ambient data (dark green dashed line) and for OFR data (brown line) are added to the figure as well as quantiles of OFR data (light brown dashed lines).