# Peer review of "Phase state of ambient aerosol linked with water uptake and chemical aging in the Southeastern US"

_Atmospheric Chemistry and Physics, 2016_

## Referee Comment (RC1) · Anonymous Referee #1 · 10 Jun 2016

This manuscript describes the phase state of ambient organic dominated particles in the Southeast US during the SOAS campaign. The phase state is using the aerosol bounce technique. The phase state was measured concurrently with hygroscopicity measurements using a HTDMA and chemical composition using an AMS. The ambient particles are further aged using an oxidation flow reactor and the phase state observed. The manuscript is well-written and I support publication in ACP after the following comments are addressed.

Comments:

1) The authors mention that the most interesting effects on phase state occur between 30 -70% RH (for isoprene). However, 75% of the time the ambient RH was above 70% RH. Therefore, the authors are not measuring the ambient phase state of the particles. How can the authors state that at ambient RH the particles will be liquid, if these measurements were never made?

2) The data in Figure 3 do not reach zero. Again, how can the authors state that the ambient particles are liquid, if they are bouncing during impaction?

3) How can the authors rue out the possibility of two particle types, one solid phase (bouncing) and one liquid phase (sticking) vs. one particle type of intermediate phase (50% bouncing)?

4) Page 4 line 25-27: This range and phase transition is true only for a homogeneous distribution. How do the authors know this is the case in the ambient? Bateman et al., 2015 published a relationship between viscosity of sucrose particles and bounce fraction.

5) Page 11 line 10-12: Any correlation with the AMS collection efficiency? Was the collection efficiency on the same order predicted by the bounce fraction? Why or Why not?

Minor Comments:

1) Page 2 Line24 -26: Perhaps Shrivasta et al. 2013 and Zaveri et al. 2014 or Scott et al 2015 have used this in their modeling. If not, they might have 2015 or 2016 papers in which these effects are included.

2) Page 3 line 5: I believe Bateman et al. 2016 used terms such as liquid particles dominate, but they are not "always" liquid.

3) Page 4 line 11: Is eight seconds enough time for kinetically limited particle to uptake water? Berkemier has some calculations and timescales for this.

4) Page 4 lines 19 – 21: I have no idea what this means? What do the RH sensors and the CPC offsets have to do with each other? Please clarify this sentence.

5) Page 5 line 29-20: how any points were lost from this analysis? How was the external mixture determined?

6) Page 6 Equations 2 – 4: How much will uncertainties in the calculation of these quantities effect the conclusion the authors draw? The authors should comment on this in the manuscript. Technical Correction:

1) Page 3 line 16: "physical phase state" remove either physical or phase

2) Page 11 line 11: replace bounciness with a term that describes physical state i.e. viscosity, semi-solidness

1       Bateman, A. P., Bertram, A. K. & Martin, S. T. Hygroscopic influence on the semisolid-to-liquid transition of secondary organic materials. *J. Phys. Chem. A* **119**, 4386-4395 (2015).
2       Scott, C. E. *et al.* Impact of gas-to-particle partitioning approaches on the simulated radiative effects of biogenic secondary organic aerosol. *Atmos. Chem. Phys.* **15**, 12989-13001 (2015).
3       Shrivastava, M. *et al.* Implications of low volatility soa and gas-phase fragmentation reactions on soa loadings and their spatial and temporal evolution in the atmosphere. *J. Geophys. Res. Atmos.* **118**, 3328-3342 (2013).
4       Zaveri, R. A., Easter, R. C., Shilling, J. E. & Seinfeld, J. H. Modeling kinetic partitioning of secondary organic aerosol and size distribution dynamics: Representing effects of volatility, phase state, and particle-phase reaction. *Atmos. Chem. Phys.* **14**, 5153-5181 (2014).

---

## Referee Comment (RC2) · Anonymous Referee #2 · 7 Jul 2016

The paper by Pajunoja et al. investigates the phase state of ambient particles in the Southeastern US using an aerosol bounce instrument. The work shows that ambient particles in this region are mostly in the liquid state. In addition further analysis shows that the phase state is related to the hygroscopicity of the particles. The paper is very well written, the analysis is excellent, and the results are important for modelling aerosol formation and growth in the atmosphere. I highly recommend this paper for publication after the authors have had a chance to address the following comments.

Abstract, line 26-28. What sampling techniques are you referring to here? I don't think this was discussed anywhere in the main document. More specifics somewhere in the document would be useful to the reader.

[Figure]

Page 2, line 31-32. The authors state: "Saukko et al. 2012, showed that the increasing O:C of SOA particles decreases the particle liquefying RH". After reading this sentence I went back and looked at the abstract for Saukko et al. 2012. In the abstract Saukko et al. state "in the majority of cases the bounce behavior of the various SOA systems did not show correlation with the particle O:C." These two statements sound contradictory. Please clarify.

Please define the terms in equation 5. Also, why is equation 5 needed? Can't the OH exposure be calculated directly from the real-time decay of CO?

Figure 4c. I wonder if the occurrence of liquid-liquid phase separation (LLPS) in the particles is influencing the particle bounce. LLPS in particles containing organic and inorganic material is expected to occur at O:C values less than approximately 0.7 (very roughly). In Figure 4c, when the O:C is less than 0.7 significant bounce is observed even at high RH, which is when LLPS is expected. On the other hand, when the O:C is roughly 0.7 and greater, the bounce is significantly reduced, which is when LLPS is not expected.

---

## Author Comment (AC1) · 16 Aug 2016

**REVIEWER #1**

We thank the referee #1 for the positive assessment and helpful comments. Following are the comments raised (in *red and italics*), and our responses in plain text.

*This manuscript describes the phase state of ambient organic dominated particles in the Southeast US during the SOAS campaign. The phase state is using the aerosol bounce technique. The phase state was measured concurrently with hygroscopicity measurements using a HTDMA and chemical composition using an AMS. The ambient particles are further aged using an oxidation flow reactor and the phase state observed. The manuscript is well-written and I support publication in ACP after the following comments are addressed.*

*Comments:*
*1) The authors mention that the most interesting effects on phase state occur between 30 -70% RH (for isoprene). However, 75% of the time the ambient RH was above 70% RH. Therefore, the authors are not measuring the ambient phase state of the particles. How can the authors state that at ambient RH the particles will be liquid, if these measurements were never made?*

Typical BF curve of amorphous SOA particles with RH is a decreasing sigmoidal curve (see e.g. Saukko et al. 2012 Fig. 5., Pajunoja et al. 2015 Fig. 2., Bateman et al. 2016 Fig 2.), and this information can be used when extrapolating the bounce measurement results to cover the whole RH range. In this study we wanted to study the effect of O:C and hygroscopicity on BF. During the first days of the campaign we performed some measurements at RH > 75% and as the BF values were very low in general, the differences in BF were small. As we focused on studying the effect of O:C and hytgroscopicity on BF, we lowered the measured RH range to 30-70% to see clearer variations in bounce. However, the reviewer is correct, we do not have bounce measurements reported in the range of RH>70% in the manuscript. Hence, we have reworded the manuscript and we now conclude that (Page 1 line 24, Page 9 line 2, and Page 11 line 31) "our results indicate that organic dominated particles stay mostly liquid in the atmospheric conditions in the Southeast US".

*2) The data in Figure 3 do not reach zero. Again, how can the authors state that the ambient particles are liquid, if they are bouncing during impaction?*

As mentioned above, the typical bounce curve for SOA particles is previously studied and based on the studies we can extend our results to higher RH range. Based on e.g. Bateman et al. (2015), the viscocity of SOA particles having approx. BF < 0.4 is already low enough to assume liquid like behavior of particles when atmospheric processes are considered.

*3) How can the authors rue out the possibility of two particle types, one solid phase (bouncing) and one liquid phase (sticking) vs. one particle type of intermediate phase (50% bouncing)?*

Externally mixed particle population containing both liquid and semisolid particles with different chemical compositions should have different hygroscopicity resulting multimodal GF-PDFs. Time periods when GF-PDFs were multimodal were excluded from the analysis and only time periods representing the internally mixed particles (single mode GF-PDFs) were included in the analysis (mentioned in Page 5 line 28-).

*4) Page 4 line 25-27: This range and phase transition is true only for a homogeneous distribution. How do the authors know this is the case in the ambient? Bateman et al., 2015 published a relationship between viscosity of sucrose particles and bounce fraction.*

Externally mixed particles were filtered with the method explained above. Based on the dataset used in this study, we cannot distinguish liquid-semisolid phase separation from fully mixed semisolid particles which have equal BF. To avoid the dominating effect of inorganics on total BF we narrowed our analysis to include only periods where the organic mass fraction (derived from AMS analysis) was greater than 0.6. As shown in Saukko et al. (2015), within this range the effect of ammonium sulfate deliquescence is diminished and the total BF is dominated by organic fraction of the particles.

*5) Page 11 line 10-12: Any correlation with the AMS collection efficiency? Was the collection efficiency on the same order predicted by the bounce fraction? Why or Why not?*

The correlation between AMS collection efficiency and bounced fraction is very interesting topic and under investigation in near future, which will be addressed in an another technique paper on AMS quantification due to its complexity. It is good to keep in mind that when the particles are dried before sampling inlet with Nafion dryer to achieve RH<30% for quantification purpose and impacted to the heated AMS vaporizer. In the vacuum system of AMS, the particles are in low pressure and thus should be further dried. As the viscosity is a function of RH and T, the actual viscosity in the AMS vaporizer may be very different compared to viscosities in the impactor conditions. Thermal decomposition of particle on the AMS vaporizer may also influence the CE, which will not be seen in the impactor condition.

*Minor Comments:*

*1) Page 2 Line24 -26: Perhaps Shrivasta et al. 2013 and Zaveri et al. 2014 or Scott et al 2015 have used this in their modeling. If not, they might have 2015 or 2016 papers in which these effects are included.*

We thank reviewer for the good references. Those are now included to the text as follow (Page 2 line 19-21): "Zaveri et al. (2014) took the lower diffusion rates into account in modeling kinetic partitioning and size distribution kinetics. Recently, the kinetic approach where organic material condenses according to the surface area rather than organic mass of the particles has been tested in the aerosol microphysical models (Shrivastava et al. 2013; Scott et al. 2015). We have also modified the text (Page 2 line 28).

*2) Page 3 line 5: I believe Bateman et al. 2016 used terms such as liquid particles dominate, but they are not "always" liquid.*

We have modified the text accordingly.

*3) Page 4 line 11: Is eight seconds enough time for kinetically limited particle to uptake water? Berkemier has some calculations and timescales for this.*

This is a relevant question! For condensation/absorption of big organic molecules the timescale might be too short. Based on Pajunoja et al. (2015), water uptake of 100-120nm semisolid SOA-particles did not increase when the humidification time in HTDMA was increased. Thus, 8 seconds should be long enough for small water

molecules to diffuse. It should be also noted that Stokes-Einstein relation is not valid for water molecules diffusing inside the organic matrix and the diffusion coefficients are clearly higher than predicted using Stokes-Einstein (e.g. Marshall et al., 2016).

*4) Page 4 lines 19 – 21: I have no idea what this means? What do the RH sensors and the CPC offsets have to do with each other? Please clarify this sentence.*

We agree with the reviewer; the sentence is unclear. Checking the RH sensors has nothing to do with the CPCs. The text is now modified:
"Outputs of RH sensors (Vaisala, HMP-110) used in the ABI were compared to the theoretical deliquescence RH of pure ammonium sulfate by measuring its humidogram with the ABI prior and after the campaign. The offset of the two CPCs (i.e. $BF_{reference}$) instead was measured every second day, at the very least."

*5) Page 5 line 29-20: how any points were lost from this analysis? How was the external mixture determined?*

Data coverage of multimodal GF-PDFs (i.e. externally mixed cases) was less than 10%. All the HTDMA scans were analyzed and both unimodal and multimodal distributions were fitted to each scan. In most of the cases the unimodal distribution was clearly the best fit for the GF data. This was confirmed by comparing the goodness of fitting of unimodal and multimodal fits and also confirmed by eye.

*6) Page 6 Equations 2 – 4: How much will uncertainties in the calculation of these quantities effect the conclusion the authors draw? The authors should comment on this in the manuscript.*

It is true that the mixing rule calculations are sensitive to assumptions made in defining the inorganic fraction. The approach how the inorganic fractions are calculated is based on thermodynamic equilibrium model ISORROPIA (Nenes et al. 1998). If we trust AMS analysis, the composition of inorganics should be reliable. Anyhow, we have limited our analysis to periods where the mass fraction of inorganics has been lower than 40%. In the worst scenario all the inorganics would be consisted of either pure sulfuric acid or pure ammonium sulfate which is not the case since the particles have shown to be acid during the SOAS campaign. Nevertheless, we tested this scenario and it generates less than 0.04 spread (the difference between the scenarios and the reported $\kappa_{OA}$ being somewhere in between) in $\kappa_{OA}$ over 95% of the dataset. The uncertainties do

not affect the trends in $\kappa_{OA}$ but the absolute values may be slightly different. We have now added a phrase about the uncertainties in mixing rule calculations as follow (Page 10 line 5-7):

"The sensitivity of the method to calculate $\kappa_{OA}$ was also tested by varying $f_{AS}$, $f_{ABS}$ and $f_{SA}$. In more than 95% of the cases the spread in $\kappa_{OA}$ is less than 0.04 and it does not change the trends in Fig. 4d."

*Technical Correction*
*1) Page 3 line 16: "physical phase state" remove either physical or phase*

Word "phase" now removed.

*2) Page 11 line 11: replace bounciness with a term that describes physical state i.e. viscosity, semi-solidness*

Word "bounciness" is replaced with "semi-solidness".

*1 Bateman, A. P., Bertram, A. K. & Martin, S. T. Hygroscopic influence on the semisolid-to-liquid transition of secondary organic materials. J. Phys. Chem. A 119, 4386-4395 (2015).*

*2 Scott, C. E. et al. Impact of gas-to-particle partitioning approaches on the simulated radiative effects of biogenic secondary organic aerosol. Atmos. Chem. Phys. 15, 12989-13001 (2015).*

*3 Shrivastava, M. et al. Implications of low volatility soa and gas-phase fragmentation reactions on soa loadings and their spatial and temporal evolution in the atmosphere. J. Geophys. Res. Atmos. 118, 3328-3342 (2013).*

*4 Zaveri, R. A., Easter, R. C., Shilling, J. E. & Seinfeld, J. H. Modeling kinetic partitioning of secondary organic aerosol and size distribution dynamics: Representing effects of volatility, phase state, and particle-phase reaction. Atmos. Chem. Phys. 14, 5153-5181 (2014).*

References

Bateman, A. P., Bertram, A. K., and Martin, S. T.: Hygroscopic influence on the semisolid-to-liquid transition of secondary organic materials, J. Phys. Chem. A, 119(19), 4386-4395, 2015.

Bateman, A. P., Gong, Z., Liu, P., Sato, B., Cirino, G., Zhang, Y., Artaxo, P., Bertram, A. K., Manzi, A. O. and Rizzo, L. V.: Sub-micrometre particulate matter is primarily in liquid form over Amazon rainforest, Nature Geoscience, 9, 34-37, 2016.

Marshall, F. H., Miles, R. E., Song, Y. C., Ohm, P. B., Power, R. M., Reid, J. P., and Dutcher, C. S.: Diffusion and reactivity in ultraviscous aerosol and the correlation with particle viscosity. Chem. Sci., *7*(2), 1298-1308, 2016.

Nenes, A., Pandis, S. N. and Pilinis, C.: ISORROPIA: A new thermodynamic equilibrium model for multiphase multicomponent inorganic aerosols, Aquat. Geochem., 4, 123-152, 1998.

Pajunoja, A., Lambe, A. T., Hakala, J., Rastak, N., Cummings, M. J., Brogan, J. F., Hao, L., Paramonov, M., Hong, J. and Prisle, N. L.: Adsorptive uptake of water by semisolid secondary organic aerosols, Geophys. Res. Lett., 42, 3063-3068, 2015.

Saukko, E., Kuuluvainen, H., and Virtanen, A.: A method to resolve phase state of aerosol particles. Atmos. Meas. Tech., 1, 259-265, 2012.

---

## Author Comment (AC2) · 16 Aug 2016

**REVIEWER #2**

We thank the referee #2 for the positive assessment and helpful comments. Following are the comments raised (in *red and italics*), and our responses in plain text.

*The paper by Pajunoja et al. investigates the phase state of ambient particles in the Southeastern US using an aerosol bounce instrument. The work shows that ambient particles in this region are mostly in the liquid state. In addition further analysis shows that the phase state is related to the hygroscopicity of the particles. The paper is very well written, the analysis is excellent, and the results are important for modelling aerosol formation and growth in the atmosphere. I highly recommend this paper for publication after the authors have had a chance to address the following comments.*

*Abstract, line 26-28. What sampling techniques are you referring to here? I don't think this was discussed anywhere in the main document. More specifics somewhere in the document would be useful to the reader.*

By "sampling techniques" in the abstract we refer to all the aerosol sampling systems where the particles are dried to lower than ambient RH. Vast majority of the aerosol inlets used in the field and laboratory studies contains drying method, including chemical reactivity and volatility measurements. In worst case, the particle viscosity increases by three to four orders of magnitude due to drying, which may lead to dramatic changes in physical and chemical ability to interact with surrounding gas-phase. We have now added couple sentences to the main text about the issue as follow (Page 9 line 4-6):

"Such a clear difference in BF between dry and ambient RH indicates the possibility that the aerosol particles may undergo phase transition from liquid to semisolid when dried in any sampling system. This could cause measurement error when investigating for instance evaporation/condensation, chemical reactivity or volatility."

*Page 2, line 31-32. The authors state: "Saukko et al. 2012, showed that the increasing O:C of SOA particles decreases the particle liquefying RH". After reading this sentence I went back and looked at the abstract for Saukko et al. 2012. In the abstract Saukko et al. state "in the majority of cases the bounce behavior of the various SOA*

*systems did not show correlation with the particle O:C." These two statements sound contradictory. Please clarify.*

The reviewer is correct, Saukko et al. (2012) is not the best reference to this specific questions since due to the methodological restrictions at that time the RH was limited to lower values where the differences in bounce where small regardless of varying O:C. Thus, we have now removed the sentence and used more recent reference (Pajunoja et al. 2015) showing clearly the effect of O:C on particle liquefying RH.

*Please define the terms in equation 5. Also, why is equation 5 needed? Can't the OH exposure be calculated directly from the real-time decay of CO?*

Very good point! The previous analysis rested on equation (5) due to poor data coverage in CO data. Recently the CO data has been re-analyzed and the data coverage was improved. Thus, the revised OH exposure is now calculated directly from the real-time decay of CO as the reviewer #2 suggested, and the equation (5) is removed from the manuscript. The methods are compared comprehensively in Hu et al. (2016) (see Supplementary Information Fig. S4 therein).
Due to the slight changes in OH exposure values, Figure 5a is reproduced with the new values. As can be seen, the overall trend in colors did not change even the absolute values of OH exposure changed slightly.

During the re-analysis of the OH data we noticed also that the Fig. 5c had been plotted with incomplete O:C data. Thus, the Fig. 5c is now replaced by the revised one. This did not affect the conclusions.

*Figure 4c. I wonder if the occurrence of liquid-liquid phase separation (LLPS) in the particles is influencing the particle bounce. LLPS in particles containing organic and inorganic material is expected to occur at O:C values less than approximately 0.7 (very roughly). In Figure 4c, when the O:C is less than 0.7 significant bounce is observed even at high RH, which is when LLPS is expected. On the other hand, when the O:C is roughly 0.7 and greater, the bounce is significantly reduced, which is when LLPS is not expected.*

This is very interesting question. Based on Saukko et al. (2015) it is not possible to distinguish mixed and phase separated cases by bounce measurement unless both, the humidification and drying cycles, are measured. If both cycles are measured it is

possible to detect the hysteresis behavior in bounce curves, if the inorganic and organic phases are separated and if the organic fraction is not too high. Hence, based on the dataset shown in this study, we cannot distinguish the phase separation and fully mixed particles from the data and it is possible that the bounce is affected by the particle structure. Anyhow, to minimize the effect of inorganic fraction on total BF we narrowed our analysis to include only periods where the organic mass fraction (derived from AMS analysis) was greater than 0.6. We would like to also note, that the lab results (e.g. Pajunoja et al., 2015) indicates that at humidified conditions the bounce of SOA particles with no inorganic fraction clearly depends on O:C of organic material. Hence we believe that the behavior presented in Fig. 4. and Fig. 5. are dominated by the water uptake of particles and the possible structural effects play a minor role.

References

Hu, W., Palm, B. B., Day, D. A., Campuzano-Jost, P., Krechmer, J. E., Peng, Z., de Sá, S. S., Martin, S. T., Alexander, M. L., Baumann, K., Hacker, L., Kiendler-Scharr, A., Koss, A. R., de Gouw, J. A., Goldstein, A. H., Seco, R., Sjostedt, S. J., Park, J.-H., Guenther, A. B., Kim, S., Canonaco, F., Prévôt, André. S. H., Brune, W. H., and Jimenez, J. L.: Volatility and lifetime against OH heterogeneous reaction of ambient Isoprene Epoxydiols-Derived Secondary Organic Aerosol (IEPOX-SOA), Atmos. Chem. Phys. Discuss., doi:10.5194/acp-2016-418, in review, 2016.

Pajunoja, A., Lambe, A. T., Hakala, J., Rastak, N., Cummings, M. J., Brogan, J. F., Hao, L., Paramonov, M., Hong, J. and Prisle, N. L.: Adsorptive uptake of water by semisolid secondary organic aerosols, Geophys. Res. Lett., 42, 3063-3068, 2015.

Saukko, E., Zorn, S., Kuwata, M., Keskinen, J., and Virtanen A., Phase State and Deliquescence Hysteresis of Ammonium-Sulfate-Seeded Secondary Organic Aerosol, Aer. Sci. Tech., 49, 531-537, 10.1080/02786826.2015.1050085, 2015.

---

## Author Response (AR2)

We thank the Co-editor for the comments. Following are the comments raised (in *__red__ and italics*), and our responses in plain text.

*Comments to the Author:*
*Thanks for response and revised manuscript. Even though the authors did good job in responding to referees, some of the important points raised by referees were not adequately implemented in the revised manuscript. Please consider the below points and implement them in the main text:*

*- Referee 1, 3rd major comment: Possibility of two particle types, one solid phase (bouncing) and one liquid phase (sticking) vs. one particle type of intermediate phase (50% bouncing). This point should be discussed (at least mentioned).*

- We have now added a sentence discussing this issue (Page 6, lines 5-7).

*- Referee 1, 3rd minor comment: This is an important point to be justified in the paper. For example, Shiraiwa et al., PCCP, 15, 11441, 2013 has shown that equilibration timescale of hygroscopic growth should be less than mili-seconds for SOA (Fig. 4b), which can be used to support author's argument.*

- We thank co-editor for this relevant reference. It has now been added to the manuscript with short discussion (Page 4, lines 15-17).

*- Referee 2: Potential occurrence of phase separation and its effect on particle bounce is potentially important and interesting point to be discussed. I suggest adding at least one paragraph discussing this issue.*

- This has now been discussed in the revised manuscript (Page 11, lines 14-19).

*- Abstract, L26: Please remove the word "instantaneous". Liquid phase does not always mean instantaneous partitioning. Timescales of partitioning depend not only on phase state but also on many other factors such as particle size/concentrations, etc.*

- the word has been removed from the abstract (Page 1, line 26).

[revised manuscript text omitted]